# A Systematic Review of Anatomical Variations of the Inferior Thyroid Artery: Clinical and Surgical Considerations

**DOI:** 10.3390/diagnostics15151858

**Published:** 2025-07-23

**Authors:** Alejandro Bruna-Mejias, Carla Pérez-Farías, Tamara Prieto-Heredia, Fernando Vergara-Vargas, Josefina Martínez-Cid, Juan Sanchis-Gimeno, Sary Afandi-Rebolledo, Iván Valdés-Orrego, Pablo Nova-Baeza, Alejandra Suazo-Santibáñez, Juan José Valenzuela-Fuenzalida, Mathias Orellana-Donoso

**Affiliations:** 1Departamento de Ciencias y Geografía, Facultad de Ciencias Naturales y Exactas, Universidad de Playa Ancha, Valparaíso 2360072, Chile; alejandro.bruna@upla.cl; 2Departamento de Morfología, Facultad de Medicina, Universidad Andrés Bello, Santiago 8370227, Chile; c.prezfaras@uandresbello.edu (C.P.-F.); t.prietoheredia@uandresbello.edu (T.P.-H.); f.vergaravargas@uandresbello.edu (F.V.-V.); j.martnezcid@uandresbello.edu (J.M.-C.); pablo.nova@usach.cl (P.N.-B.); 3Giaval Research Group, Faculty of Medicine, University of Valencia, 46010 València, Spain; juan.sanchis@uv.es; 4Facultad de Medicina y Ciencia, Universidad San Sebastián, Lota 2465, Santiago 7510157, Chile; sary.afandi@uss.cl; 5Facultad de Ciencias de la Salud, Universidad Autónoma de Chile, Santiago 8910060, Chile; ivan.valdes.orrego@gmail.com; 6Department of Morphology and Function, Faculty of Health Sciences, Universidad de las Américas, Santiago 8370040, Chile; alej.suazo@gmail.com; 7Departamento de Ciencias Química y Biológicas Facultad de Ciencias de la Salud, Universidad Bernardo O’Higgins, Santiago 8370993, Chile; juan.kine.2015@gmail.com; 8Escuela de Medicina, Universidad Finis Terrae, Santiago 7501015, Chile

**Keywords:** inferior thyroid artery, anatomical variations, clinical anatomy, surgical anatomy

## Abstract

**Background/Objectives:** The inferior thyroid artery (ITA) is an essential component of the thyroid gland’s vasculature, with significant clinical and surgical implications due to its anatomical variability. This systematic review aimed to describe the prevalence of ITA anatomical variants and their association with clinical conditions or surgical implications. **Methods:** A comprehensive search was conducted in MEDLINE, Web of Science, Google Scholar, CINAHL, Scopus, and EMBASE on 20 November 2025. Eligibility criteria included studies reporting on the presence of ITA variants and their correlation with pathologies. Two authors independently screened the literature, extracted data, and assessed methodological quality using the AQUA and JBI tools. **Results:** Of the 2647 articles identified, 19 studies involving 1118 subjects/cadavers were included. Variations in ITA origin, absence, and additional arteries were reported, with the most common variant being direct origin from the subclavian artery. Clinically, these variations were associated with increased risk of intraoperative hemorrhage, potential nerve damage, and challenges in preoperative planning, particularly during thyroidectomy and other neck procedures. **Conclusions:** Understanding the anatomical diversity of the ITA is crucial for reducing surgical risks and improving patient outcomes. The review highlighted the need for more standardized research protocols and comprehensive data reporting to enhance the quality of evidence in this domain. Preoperative imaging and thorough anatomical assessments tailored to individual patient profiles, considering ethnic and gender-related differences, are essential for safe surgical interventions in the thyroid region.

## 1. Introduction

The inferior thyroid artery (ITA) is a vital component of the thyroid gland’s vascular supply, responsible for nourishing its lateral and inferior regions as well as the surrounding prethyroid muscles. Typically, the ITA is described as a branch of the thyrocervical trunk, and its anatomical significance is underscored by its close relationship with the recurrent laryngeal nerve (RLN), which is crucial for the vagal innervation of the thyroid gland.

From a surgical perspective, the ITA serves as a contentious landmark for accessing the cervical region, primarily due to its intricate connections with critical structures such as the RLN. This complexity is aggravated by the anatomical variability observed in the ITA among different populations. Toni et al. (2005) [1] emphasize that variations in the origin, branching patterns, and anatomical relationships of the ITA can significantly impact surgical outcomes. Their meta-analysis highlights the diversity of these anatomical variations across different ethnic groups, which are crucial for understanding potential complications during thyroid surgeries. Notably, the literature indicates that approximately 1% of these variations are numerical, with the ITA’s origin typically arising from the thyrocervical trunk in 90–95% of cases, depending on ethnic factors. However, it may also originate from the subclavian artery and, in rare instances, from the vertebral or common carotid arteries (Branca et al., 2022) [2]. Recent studies, such as those by Bunea et al. (2023) [3], have highlighted that approximately 31.48% of ITAs may originate directly from the subclavian artery, with significant variations observed based on the side of the body and gender.

The variability of the ITA anatomy is particularly significant when considering surgical implications, as discrepancies in its presence and lateralization have been documented among different ethnic groups. Additionally, the relationship between the ITA and the RLN is variable, with studies reporting that in 52.08% of cases, the artery is positioned anterior to the nerve (Noussios et al., 2020) [4]. Such variations can directly influence the risk of nerve injury during thyroidectomy, a procedure that carries a complication rate of 2–5% for RLN damage in patients without nerve variations. These complications can lead to serious outcomes, including voice and swallowing difficulties, underscoring the need for careful surgical techniques to accurately identify the ITA as a reference point for locating the RLN (Branca et al., 2022) [2].

A thorough understanding of the thyroid gland’s vascular anatomy, particularly the ITA and its variations, is essential for effective surgical planning. Toni et al. (2005) [1] suggest that comprehensive preoperative evaluation and imaging are critical for anticipating anatomical variations and reducing surgical risks. Surgeons must be well-informed about these anatomical details to minimize risks associated with thyroid surgery. As such, the compilation and dissemination of data regarding anatomical variants are crucial from both clinical and surgical perspectives. Modern imaging modalities, such as CT angiography, can aid in identifying anatomical variants, thus enhancing surgical preparedness as well (Caruso et al., 2000; Francis et al., 2014; Branca et al., 2022; Toni et al., 2005; Ray et al., 2012; Bunea et al., 2023) [1,2,3,5,6,7].

Furthermore, the presence of anomalous vessels such as the thyroidea ima artery (TIA) can further complicate surgical interventions, necessitating detailed preoperative imaging and planning to avoid inadvertent vascular injury (Ray et al., 2012) [7]. Consequently, understanding these anatomical variants is critical for reducing surgical risks and improving patient outcomes.

While previous reviews have established foundational knowledge regarding anatomical variations of the ITA, this review expands on that knowledge by providing a comprehensive analysis of recent findings, including the prevalence of various ITA anatomical variants and their clinical implications. Specifically, this review offers a detailed examination of the prevalence rates of different ITA variants, underscoring their significance in surgical practice and highlighting the need for individualized preoperative planning to mitigate risks associated with these variations. Moreover, this work emphasizes the importance of standardized reporting and methodological rigor in future studies, thereby enhancing the quality of evidence available in the field. By addressing these critical aspects, this review not only adds valuable information to the existing literature but also sets a precedent for future research aimed at improving surgical outcomes and patient safety in thyroid surgeries.

Therefore, this review aims to comprehensively investigate the morphology of the ITA, considering its anatomical variants and exploring their implications for surgical management. By understanding these factors, surgeons can improve outcomes and reduce the likelihood of complications during thyroid surgeries.

## 2. Methods

### 2.1. Protocol

This systematic review and meta-analysis were performed and reported according to the Preferred Reporting Items for Systematic Reviews and Meta-Analyses (PRISMA) statement (Page et al., 2021) [8]. Also, this systematic review was registered in the Open Science Framework (OSF) and has the following DOI:10.17605/OSF.IO/JPMUY.

### 2.2. Eligibility Criteria

Studies on the presence of ITA variants and their association with any clinical condition were considered eligible for inclusion if the following criteria were fulfilled: (1) population: sample of dissections or images of the ITA; (2) outcomes: ITA prevalence, variants, and their correlation with any condition or surgical utility. Additionally, anatomical variants such as variations in the origin, trajectory and branching pattern were sought, classified and described based on normal anatomy and classifications proposed in the literature; (3) studies: this systematic review included research articles, research reports, or original research published in English or Spanish in peer-reviewed journals and indexed in some of the databases reviewed. Conversely, the exclusion criteria were as follows: (1) population: animal studies; studies: letters to the editor or comments.

### 2.3. Electronic Search

We systematically searched MEDLINE (via PubMed), Web of Science, Google Scholar, the Cumulative Index to Nursing and Allied Health Literature (CINAHL), Scopus, and EMBASE until 20 January 2025.

The search strategy included a combination of the following terms: “inferior thyroid artery” (No Mesh) “variants” (No Mesh) using the Boolean connectors “AND”, “OR” and “NOT” (Figure 1). The search strategies for each database are available in the Appendix A.

### 2.4. Study Selection

Two authors (CP and TP) independently screened the titles and abstracts of references retrieved from the searches. We obtained the full text for references that either author considered to be potentially relevant. We involved a third reviewer (FV) if consensus could not be reached.

### 2.5. Data Collection Process

Three authors (CP, TP and FV) independently extracted data on the outcomes of each study. The following data were extracted from the original reports: (i) authors and year of publication, (ii) type of study, (iii) type of technique used to observe the ITA, (iv) sample characteristics (sample size, age, country, ethnicity, gender, laterality), (v) ITA’s variant prevalence and morphological characteristics, (vi) clinical/surgical implications.

### 2.6. Methodological Quality Assessment of the Included Studies

The quality assessment of retrospective and prospective observational studies was performed using the methodological quality assurance for anatomical studies (AQUA) tool proposed by the International Evidence-Based Anatomy (IEBA) Working Group based on five domains (Table 1), which is a methodological framework designed to standardize the reporting of anatomical studies that ensures rigor, transparency, and reproducibility. Developed through a Delphi consensus process involving global anatomical experts, it addresses key domains critical to minimizing bias and enhancing clinical relevance. The checklist mandates explicit documentation of demographic characteristics (e.g., age, sex, medical history) and study design fundamentals, including prospective/retrospective approaches and anatomical methodologies (cadaveric, imaging, or intraoperative). It emphasizes sample size justification through statistical power analysis and requires clear definitions of anatomical reference standards (e.g., variations, classifications) to ensure comparability across studies. The methodological quality of included studies was assessed using the Anatomical Quality Assurance (AQUA) Checklist, which prioritizes transparency through explicit descriptions of techniques, instruments, and measurement protocols to ensure replicability (Tomaszewski et al., 2017; Iwanaga et al., 2025) [9,10]. This tool evaluates diagnostic and intervention clarity, including drug dosages, procedural steps, and post-intervention outcomes (e.g., adverse events) while mandating ethical compliance with guidelines such as the Helsinki Declaration. For case reports, the Joanna Briggs Institute (JBI) critical appraisal checklist (Table 2) was applied to assess methodological rigor, validity, and clinical relevance, emphasizing demographic clarity (age, sex, medical history), chronological clinical timelines, and diagnostic method alignment with gold standards (Gagnier et al., 2013; Ray et al., 2012) [7,11]. Two reviewers (CPF and TPH) independently conducted data extraction and quality assessments, with discrepancies resolved by a third reviewer (FVV). Studies were classified by bias risk using predefined criteria: low risk (>70% “yes” responses), moderate risk (50–69%), or high risk (<49%). Statistical analyses, including Student’s *t*-tests and chi-square tests (*p* < 0.05), were applied to variables such as anatomical variant prevalence, with cluster analysis identifying safer surgical zones (e.g., lingual foramina positioning relative to the genial tubercle) (Archibong et al., 2023; Bunea et al., 2023) [3,12].

## 3. Results

### 3.1. Included Articles

The search resulted in a total of 2647 articles from different databases that met the criteria and search terms established by the research team. The filter was applied to the articles’ titles and/or abstracts in the consulted databases. Following the removal of duplicate records, such those eliminated by automated tools or for other reasons, such as documents that did not address the ITA’s anatomy, animal studies, and gray literature, were excluded. In total, 240 full-text articles were evaluated for eligibility for inclusion. Next, 221 studies were excluded because their primary and secondary results did not match those of this review or because they did not meet the established criteria for good data extraction, resulting in 19 articles included for analysis with a total of 1118 subjects/cadavers (Figure 1).

### 3.2. Characteristics of the Included Studies and the Study Population

Within the 19 included studies, consisting of 16 case reports and 3 cross-sectional studies, a total of 1118 subjects/cadavers were recorded (Table 3). Among the characteristics observed in the compiled studies about study subjects, the gender distribution of the 1118 subjects/patients recorded includes a total of 439 (39.27%) females, 549 (49.11%) males. Additionally, the ethnicities described include a total of 110 (9.83%) Caucasians, 1 (0.09%) South Asian, and 1007 (90.01%) subjects with unspecified ethnicity. On the other hand, regarding the techniques used for the study of ITA, the utilization of coronary computed tomography angiography (CTA) is described in 2 studies, cadaveric dissection in 14 studies, and thyroidectomy in 3 studies.

### 3.3. Methodological Quality of the Included Studies

Among the included studies, there were 16 case reports, 1 clinical study, 1 clinical observational study, and 1 research article (see Table 3) using the AQUA or the JBI tool (Table 1 and Table 2). Of the case reports evaluated by JBI, 18.75% exhibited a low risk of bias, 12.5% a moderate risk, and 68.75% a high risk of bias. The articles categorized as having a low risk of bias include those by Yang et al. (2024) [26], Ngo Nyeki et al. (2016) [21], and Mariolis-Sapsakos et al. (2014) [18], all of which utilized thyroidectomy techniques and met most of the JBI criteria. In contrast, the articles with a moderate risk of bias are represented by Bunea et al. (2023) [3], who employed computed tomography angiography (CTA) techniques, and Lovasova et al. (2017) [17], who utilized cadaveric methods. Notably, the predominant number of articles exhibiting a high risk of bias pertain to cadaveric studies. This necessitates the assignment of “Not Applicable” or “No” for criteria 3 and 6, as these criteria are not relevant in the context of cadaver studies. Similarly, criterion 4 also faces challenges due to the same reasoning. It is important to emphasize that while these latter articles may not align with the reliability standards set forth by the JBI criteria, this discrepancy arises from the inherent nature of cadaveric studies rather than indicating irrelevance or unreliability of the findings described, as they still contribute valuable insights despite not being part of an in vivo study group (see Table 4).

The studies considered in the analysis conducted using the AQUA tool include one case report (Ray et al., 2012) [7] and two cross-sectional studies (Özgüner et al., 2014) [23] (Esen et al., 2018) [14], all of which generally present a low risk of bias, particularly highlighting the items “study design” and “descriptive anatomy,” both of which demonstrate a 100% low bias risk (Table 4, Figure 2).

However, mild deviations in bias are observed in domains 1, 3, and 5. Correspondingly, the only article that presents conflicts related to objective and subject characteristics was conducted by Ray et al. (2012) [7], in response to question 2. The population used in this study is grossly inadequate for preliminary characterization of such a vast population as Karnataka, India.

Concerning methodology characterization, two of the three studies (Ray et al., 2012 [7]; Özgüner et al., 2014) [23] omit all relevant information regarding the expertise of the individuals involved in response to question 11. Furthermore, in response to question 15, only one article (Esen et al., 2018) [14] provided bias related to the nature of the study itself, which was retrospective in design, and consequently, an absence of a gold standard for performing comparisons.

Finally, regarding the reporting of results, only one study failed to report part of the results accurately due to a lack of fundamental data such as gender, side, and age (Ray et al., 2012) [7], contributing to a negative response to question 22 (Moola et al., 2017) [29].

### 3.4. Inferior Thyroid Artery’s Morphology

The studies and case reports reviewed reveal several interesting variations in the ITA, including cases of absence, differences in origin, and other anatomical variants. In some instances, the ITA was missing on one side, with compensation from other arteries Table 5). For example, Sherman et al. (2003) [24] found a male with no left ITA, where branches from the right ITA took over the supply. Similarly, González-Castillo et al. (2018) [15] documented a female with situs inversus totalis who lacked a right ITA but had two left ITAs making up for it. Jelev et al. (2001) [16] reported a male with an absent right ITA, compensated for by an enlarged STA. Moriggl et al. (1996) [19] described a female cadaver missing ITAs on both sides, with a branch from the left internal thoracic artery providing the necessary blood supply. Another study by Novakov et al. (2023) [22] highlighted a female with no ITAs, where large STAs and an ITA from the brachiocephalic trunk compensated for the absence.

The origin of the ITA can vary significantly; Ray et al. (2012) [7] found an ITA originating from the right CCA in one case, while Lovasova et al. (2017) [17] noted additional arteries including an MTA from the CCA. Esen et al. (2018) [14] also reported cases where ITAs originated from the brachiocephalic trunk, CCA, and aortic arch. In some cases, additional arteries like the thymothyroid artery were found, as reported by Novakov et al. (2023) [22]. In another instance, Cigali et al. (2008) [13] identified an accessory ITA from the left suprascapular artery. Westrych et al. (2022) [25] noted a unique scenario where a common trunk for the internal thoracic artery, thyrocervical trunk, and ITA originated from the left subclavian artery.

Gender and side differences also play a role in the variations of ITA. Bunea et al. (2023) [3] analyzed arteries in men and women, finding that some ITAs originated directly from the subclavian artery, with variations between the right and left sides. In essence, these variations in the ITA’s morphology highlight the importance of understanding anatomical diversity, which is crucial for surgical planning and minimizing risks during procedures involving the thyroid gland.

### 3.5. Clinical Considerations

From the 19 reviewed studies, regardless of their nature and subject of study, the vast majority indicate, from a subjective standpoint, the implications of the previously mentioned anatomical variants. These studies do not employ objective tools to validate the reported information, whether regarding surgical, imaging, ethnic, or gender-related considerations.

In the surgical field, the literature describes potential implications as consequences of variations in the course and branching pattern of the inferior thyroid artery (ITA), its origin, and its associated structures, among others. Concerning trajectory and origin variations, Moriggl et al. (1996) [19], Ray et al. (2012) [7], Yilmaz et al. (1993) [27], and Yohannan et al. (2019) [28] emphasize the need to consider ITA vulnerability in surgical interventions in the anterior cervical region, such as tracheotomy or laryngeal transplants. They highlight procedural variations, such as the IMA variant, due to its superficial trajectory over the trachea. Moriggl et al. (1996) [19] further notes that such variants pose a high risk of uncontrollable hemorrhage, given the challenges associated with locating, identifying, and assessing the course and size of these variations. Therefore, extreme caution is advised when accessing the suprasternal fossa or performing procedures such as those mentioned above. In line with these concerns, Ngo Nyeki et al. (2016) [21] reports similar risks due to aberrant ITA trajectories even in the absence of origin variations.

Regarding the branching pattern of the ITA and its associated variations, Ray et al. (2012) [7] underscores the importance of preserving the arterial supply to the thyroid gland. The author highlights the need to safeguard these branches while ligating regional arteries during surgical interventions. Additionally, the study emphasizes that these relatively frequent variations should be considered in procedures such as catheterization.

Additional surgical implications related to ITA origin variations are reported by Natsis et al. (2009) [20], Esen et al. (2017) [14], and Westrych et al. (2022) [25]. These authors stress the importance of recognizing such variations, exemplifying their clinical significance with findings such as increased incidence of intraoperative hemorrhage (Esen et al., 2017) [14]. Moreover, undetected anatomical variations could lead to complications in procedures such as coronary artery bypass grafting (CABG). Westrych et al. (2022) [25] describes a specific variation where the ITA, thyrocervical trunk (TCT), and internal thoracic artery originate from a common trunk arising from the subclavian artery. Similarly, Natsis et al. (2009) [20] report a variation in which the ITA originates from the left vertebral artery.

Finally, concerning the relationship between the ITA and intrinsically associated structures in surgical practice, Archibong et al. (2023) [12], Ray et al. (2012) [7], Sherman et al. (2003) [24], and Ngo Nyeki et al. (2016) [21] jointly emphasize the anatomical relationship between the ITA and the recurrent laryngeal nerve (RLN) during thyroid surgery, underscoring the importance of avoiding nerve damage. These authors stress the necessity of considering the wide range of anatomical variations of the ITA, given their potential impact on the development of other pathologies, such as goiter. Ngo Nyeki et al. (2016) [21] specifically notes that ITA positioning can influence RLN trajectory, ultimately increasing the risk of iatrogenic injury.

Other clinical implications briefly mentioned in the reviewed literature pertain to ethnic variability in the studied population. Özgüner et al. (2014) [23], Yohannan et al. (2019) [28], and Jelev et al. (2001) [16], in a bibliographic review consistent with the findings presented, report a significant ethnic influence on anatomical variations of the ITA, reinforcing the need to consider ethnicity as a relevant factor in preoperative planning.

## 4. Discussion

This systematic review identified 2647 articles, ultimately including 19 studies involving 1118 subjects, with a notable international representation. The gender distribution was 39.27% female, 49.11% male, and 11.63% unspecified and, while ethnicity was not specified by most of the studies, 9.83% were Caucasians. The research reviewed highlighted variations in the ITA and their significant surgical implications. Studies indicated that variations in the ITA’s trajectory and branching could lead to surgical risks, including uncontrollable hemorrhage during procedures such as tracheotomy (Moriggl et al., 1996 [19]; Ray et al., 2012 [7]). Additionally, preserving arterial supply during surgical interventions was emphasized, with evidence suggesting that undetected anatomical variations could complicate procedures like coronary artery bypass grafting (Natsis et al., 2009 [20]; Esen et al., 2017) [14]. The review also noted the association between the ITA and the RLN, stressing the importance of considering anatomical variations to mitigate the risk of nerve damage (Archibong et al., 2023 [12]; Ngo Nyeki et al., 2016 [21]). Furthermore, ethnic variability was recognized as a critical factor in understanding anatomical differences, reinforcing the need for tailored preoperative planning (Özgüner et al., 2014) [23] (Table 6, Figure 3).

The ITA ultimately traces its origin back to cells primarily derived from the lateral plate mesoderm, particularly from the splanchnic layer, with substantial contributions from the ectoderm, precisely, from neural crest cells and inductive influences from the endoderm in early developmental stages, as stated by Centeno et al. (2021) [30] (Gilbert, 2000 [31]; Centeno et al., 2021 [30]; Khalid et al., 2023 [32]).

The formation of thyroid vessels, like ITA, occurs during the aortic arch branches formation accompanying the thyroid gland descent, a process that takes place between the 4th and 7th weeks of embryological development (Ngo Nyeki et al., 2016) [21] and involves the transformations and partial regression of the aortic arches (Khalid et al., 2023) [32] aiming to give rise to the thyrocervical trunk ascending from the subclavian artery, where it is ITA’s most common origin (Bhatia et al., 2005) [33]. In this context, it can be associated with subclavian artery development. On the right side, it is developed from the 4th aortic arch, which gives rise to the proximal segment of the subclavian artery, while the remaining portion originates from the 7th cervical intersegmental artery; on the left side, it is developed entirely by the 7th cervical intersegmental artery (Kau et al., 2007) [34].

Furthermore, ITA’s embryologic origin is closely related to RLN development and the parenchymal thyroid tissue, as indicated by Toni et al. (2005) [1], reflecting the coordinated evolution and nearness of vascular and neural structures in the cervical region (Ngo Nyeki et al., 2016 [21]; Toni et al., 2005 [1]). Even though this relevant data exists, studies have provided scant knowledge on the ITA’s embryology. Research on the ITA’s embryology receives less attention compared to other thyroid arteries such as the superior thyroid artery. Highlighting this issue, it is relevant to emphasize embryology research to ascertain relevant ITA’s structural relationships with the thyroid, RLN, and surrounding tissue.

### 4.1. Methodological Heterogeneity and Rigor of the Included Studies

The systematic review encompassed 16 case reports, one clinical study, one clinical observational study, and one research article (Table 1). According to the JBI assessment, 18.75% of the case reports exhibited a low risk of bias, 12.5% a moderate risk, and 68.75% a high risk of bias. Notably, the studies by Yang et al. (2024) [26], Ngo Nyeki et al. (2016) [21], and Mariolis-Sapsakos et al. (2014) [18], which employed thyroidectomy techniques, were classified as low risk, as they adhered closely to the JBI criteria. In contrast, the moderate risk articles, including Bunea et al. (2023) [3], who used computed tomography angiography (CTA), and Lovasova et al. (2017) [17], who utilized cadaveric techniques, highlighted the challenges inherent in cadaver studies, where criteria related to current clinical conditions were deemed not applicable. This high prevalence of bias in cadaver studies does not necessarily undermine their relevance, as the findings remain significant despite methodological constraints (Table 4). Furthermore, the analysis using the AQUA tool identified one case report (Ray et al., 2012) [7] and two cross-sectional studies (Özgüner et al., 2014 [23]; Esen et al., 2018 [14]) that presented a low risk of bias, particularly in “study design” and “descriptive anatomy.” However, mild deviations were noted in several domains, particularly concerning sample appropriateness and the expertise of those conducting the studies. These methodological shortcomings underscore the necessity of rigorous study design and reporting standards to enhance the reliability and applicability of findings in clinical practice (Figure 2). Moreover, except for the studies by Buena et al. (2023) [3], Esen et al. (2018) [14], and Ozgüner et al. (2014) [23], the heterogeneity in the reported outcomes and the lack of standardized objective measurements across the studies make it challenging to conduct a meta-analysis of the data.

### 4.2. ITA’s Variants Clinical Relevance

The ITA is a crucial anatomical structure in thyroid surgery, primarily due to its essential role in providing blood supply to the thyroid gland and its proximity to the recurrent laryngeal nerve (RLN). The anatomical variations of the ITA, which occur more frequently than those of the superior thyroid artery, are important for both anatomical knowledge and clinical practice, particularly in surgical contexts (Jelev et al., 2001 [16]; Moriggl et al., 1996 [19]). These variations may include deviations in the artery’s course, the presence of accessory arteries, or even the rare thyroid ima artery, which can stem from sources like the aortic arch or the thyrocervical trunk (Esen et al., 2018 [14]; Yohannan et al., 2019 [28]). Identifying these variations is critical during thyroidectomies and other neck procedures, as unforeseen arterial configurations can result in complications such as hemorrhage or RLN damage (González-Castillo et al., 2018; Ray et al., 2012) [7,15].

Advanced surgical techniques, such as the bilateral axillo-breast approach (BABA), provide enhanced visualization and accessibility to the ITA, ensuring safer surgical interventions. These methods allow for a comprehensive examination of the ITA’s pathway, aiding in the prevention of accidental injury to the artery and surrounding nerves (Tae et al., 2019 [35]; Zhang et al., 2023 [36]). The variable branching pattern of the ITA and its relationship with the RLN are particularly critical, as they facilitate the localization of the nerve during operations, thereby reducing the risk of nerve injury, which is reported in 1–6% of thyroid surgeries (Rajabian et al., 2017; Ray et al., 2012) [7,37]. Thorough preoperative evaluations, including imaging studies, are essential to identify these anatomical variations and to plan surgical interventions that minimize risks and enhance patient safety (Moriggl et al., 1996; Ngo Nyeki et al., 2016) [19,21].

On the other hand, ITA’s hemodynamic evaluation via Doppler ultrasound holds significant diagnostic and prognostic value in the follow-up of thyroid nodules. Color Doppler assessment of the ITA, particularly measurements of peak systolic velocity (PSV), enables clinicians to differentiate autoimmune thyroid diseases such as Graves’ disease from other pathologies, as demonstrated by Caruso et al. (2000) [5], who reported distinct PSV patterns in autoimmune conditions. In nodular thyroid disease, Doppler evaluation of the ITA aids in distinguishing benign from malignant nodules by identifying abnormal vascular patterns and increased intramodular blood flow, which are often associated with malignancy (Caruso et al., 2000) [5]. Furthermore, serial Doppler assessments during follow-up can monitor therapeutic responses, such as reduced vascularity post-treatment, providing real-time insights into nodule behavior. This non-invasive approach enhances risk stratification and guides clinical decision-making, particularly in pediatric populations where ultrasound surveillance is emphasized for early detection of nodular changes (Francis et al., 2014) [6]. Integrating ITA hemodynamic data with sonographic features like microcalcifications or lymph node abnormalities further improves diagnostic accuracy, underscoring the indispensability of Doppler ultrasound in comprehensive thyroid nodule management (Caruso et al., 2000; Francis et al., 2014) [5,6].

In addition, the study by González-Castillo et al. (2018) [15] reports a unique case of a female cadaver with situs inversus totalis and reports the absence of the right ITA, compensated for by two left ITAs. This finding underscores the importance of individualized anatomical assessments preoperatively, as variations in vascular supply can lead to increased risks of intraoperative hemorrhage and complications, especially in surgeries involving the thyroid gland. By documenting such rare anomalies, González-Castillo et al. provide crucial insights into the complexities of neck anatomy that can aid surgeons in planning their approaches, thereby enhancing patient safety and surgical outcomes (González-Castillo et al., 2018) [15]. Nevertheless, it is necessary to clarify if there is any correlation with the situs inversus.

Moreover, in the embryological development of the ITA lies the process in which explains the ITA’s variation morphology, this involves contributions from lateral plate mesoderm (vascular precursors), ectodermal neural crest cells (contributing to perivascular connective tissue), and endodermal signalling (guiding thyroid morphogenesis) (Gilbert, 2000 [31]; Khalid et al., 2023 [32]). This multilineage origin directly correlates with the anatomical variability quantified in our study; regarding the ITA’s origin, from the thyrocervical trunk (90.5% of cases) versus subclavian (7.5%) or carotid arteries (2%) (Bunea et al., 2023) [3] mirrors the dynamic regression of the 4^th^–7^th^ aortic arches during weeks 4–7 of gestation (Kau et al., 2007) [34]. Concerning its branching complexity, a mean 3.2 ± 0.8 branches per ITA aligns with ectodermal neural crest contributions to angioblast migration, which promote vascular branching (Centeno et al. (2001) [30]). Moreover, shared embryological origins between the ITA and recurrent laryngeal nerve RLN via neural crest cells explain 12% of cases with ITA-RLN entanglements, a key risk factor for intraoperative nerve injury (Ngo Nyeki et al., 2016) [21].

By contextualizing our morphometric data within this developmental framework, we establish why preoperative imaging must account for ITA variability—particularly in populations with high aortic arch anomalies (e.g., 4.7% aberrant right subclavian arteries)—to mitigate surgical complications (Esen et al., 2018) [14].

As noted by Triantafyllou et al. (2024) [38], a thorough grasp of the ITA’s anatomical variability can significantly improve surgical outcomes and reduce the likelihood of complications. These findings highlight the need for enhanced training and awareness among surgeons regarding the anatomical complexities associated with the ITA, reinforcing the notion that individual anatomical variations must be considered during surgical planning to optimize patient care.

With respect to limitations, this systematic review has several limitations that warrant consideration. First, most of the studies included were case reports, which inherently restrict the generalizability of the findings. Case reports often lack robust methodological designs and may not adequately represent larger populations, potentially leading to biased conclusions regarding the anatomical variations of the ITA. Secondly, some studies demonstrated a moderate risk of bias, especially those involving cadaveric dissections. This is due to the nature of the assessment tools, which rely on the number of affirmative responses and do not account for the “unclear” label in their application, thereby increasing the perceived risk of bias. The absence of applicable criteria relating to patients’ clinical conditions in cadaver studies can diminish the reliability of the findings. Additionally, many studies did not provide comprehensive demographic data, such as ethnicity and gender distribution, which could influence the observed anatomical variations. Third, while the review aimed to encompass a broad range of the literature, the search strategy may have overlooked relevant studies published in languages other than English or Spanish, thereby limiting the diversity of the included research. This could affect the comprehensiveness of the evidence regarding ITA’s variations across different populations. Finally, the variations in study methodologies and criteria for defining anatomical variants may contribute to inconsistencies in the reported findings. Standardizing the definitions and classifications used in future research would enhance the reliability and comparability of results, ultimately leading to more definitive conclusions about the clinical implications of ITA variations.

## 5. Conclusions

This systematic review highlights the diverse morphological variations of the ITA and their clinical implications. The results indicate that variations in the ITA’s origin, such as those originating from the subclavian artery, common carotid artery, and brachiocephalic trunk, occur with notable frequency. Absence of the ITA, compensated for by other arterial structures like the superior thyroid artery or internal thoracic artery, was also observed in several cases. Understanding these variations is essential for surgical planning, as they can significantly impact surgical outcomes and risk management, particularly in procedures involving the thyroid gland. These findings suggest a need to reevaluate current anatomical and surgical standards to incorporate the variability of ITA morphology. This could enhance surgical precision and minimize complications. Future research should focus on developing standardized approaches for documenting and interpreting anatomical variations, ensuring that such critical information is consistently integrated into clinical practice.

## Figures and Tables

**Figure 1 diagnostics-15-01858-f001:**
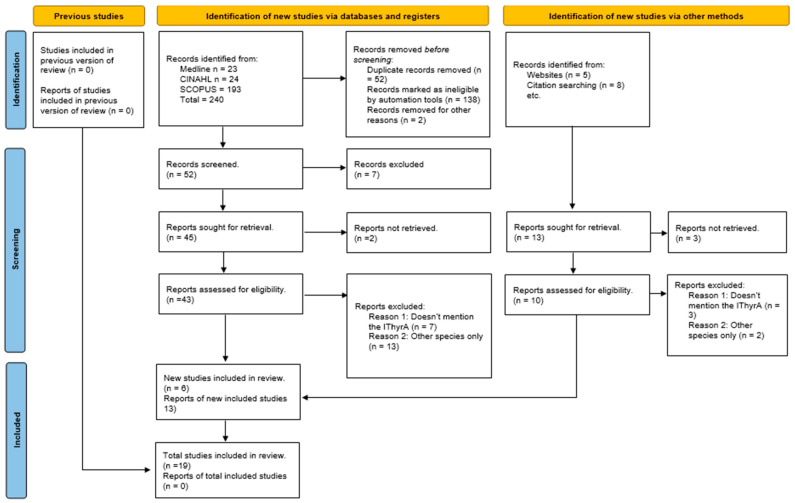
PRISMA 2020 search flow diagram for updated systematic reviews.

**Figure 2 diagnostics-15-01858-f002:**
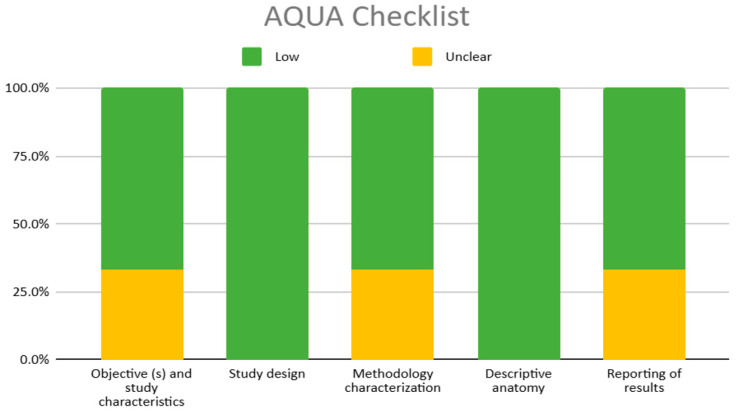
AQUA checklist risk of bias graph.

**Figure 3 diagnostics-15-01858-f003:**
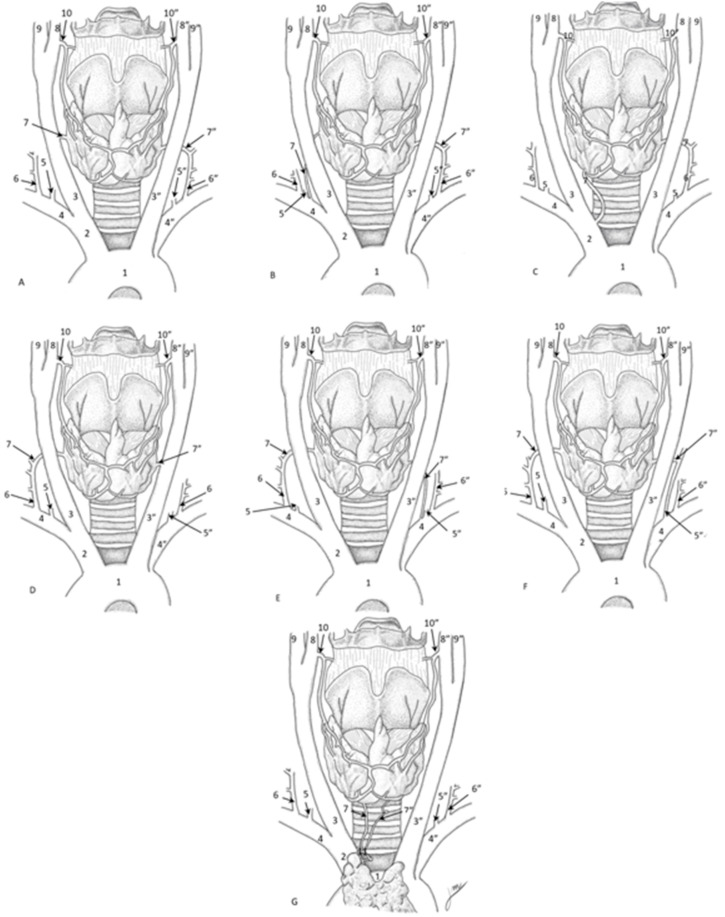
Variations in the ITA’s origin. (**A**) ITA originated from the right carotid artery. (**B**) ITA originated from the right subcalvian artery. (**C**) ITA originated from the brachiocephalic trunk. (**D**) ITA originated from the left carotid artery. (**E**) ITA originated from the left subcalvian artery. (**F**) ITA originated from the left vertebral artery. (**G**) ITA originated from a common trunk for the ITA and the superior thymic artery. 1. Aortic arch; 2. Brachiocephalic trunk; 3. Right common carotid artery; 3”. Left common carotid artery; 4. Right subclavian artery; 4”. Left subclavian artery; 5. Right vertebral artery; 5”. Left vertebral artery; 6. Right thyrocervical trunk; 6”. Left thyrocervical trunk; 7. Right inferior thyroid artery; 7”. Left inferior thyroid artery; 8. Right external carotid artery; 8”. Left external carotid artery; 9. Right internal carotid artery; 9”. Left internal carotid artery; 10. Right superior thyroid artery; 10”. Left superior thyroid artery; 11. Common trunk of the superior thymic artery.

**Table 1 diagnostics-15-01858-t001:** AQUA tool domains and question descriptions.

Domains	Questions
Domain 1: objective(s) and subject characteristics.	(1) Was (Were) the objective(s) of the study clearly defined? (yes, no, or unclear)
	(2) Was (Were) the chosen subject sample(s) and size appropriate for the objective(s) of the study? (yes, no, or unclear)
	(3) Are the baseline and demographic characteristics of the subjects (age, sex, ethnicity, healthy or diseased, etc.) appropriate and clearly defined? (yes, no, or unclear)
	(4) Could the method of subject selection have in any way introduced bias into the study? (yes, no, or unclear)
Domain 2: study design.	(5) Does the study design appropriately address the research question(s)? (yes, no, or unclear)
	(6) Were the materials used in the study appropriate for the given objective(s) of the study? (yes, no, or unclear)
	(7) Were the methods used in the study appropriate for the given objective(s) of the study? (yes, no, or unclear)
	(8) Was the study design, including methods/techniques applied in the study, widely accepted or standard in the literature? If “no”, are the novel features of the study design clearly described?
	(9) Could the study design have in any way introduced bias into the study? (yes, no, or unclear)
Domain 3: methodology characterization.	(10) Are the methods/techniques applied in the study described in enough detail for them to be reproduced? (yes, no, or unclear)
	(11) Was the specialty and the experience of the individual(s) performing each part of the study (such as cadaveric dissection or image assessment) clearly stated? (yes, no, or unclear)
	(12) Are all the materials and methods used in the study clearly described, including details of manufacturers, suppliers, etc.? (yes, no, or unclear)
	(13) Were appropriate measures taken to reduce inter- and intra-observer variability? (yes, no, or unclear)
	(14) Do the images presented in the study indicate an accurate reflection of the methods/techniques (imaging, cadaveric, intraoperative, etc.) applied in the study? (yes, no, or unclear)
	(15) Could the characterization of methods have in any way introduced bias into the study? (yes, no, or unclear)
Domain 4: descriptive anatomy.	(16) Were the anatomical definition(s) (normal anatomy, variations, classifications, etc.) clearly and accurately described? (yes, no, or unclear)
	(17) Were the outcomes and parameters assessed in the study (variation, length, diameter, etc.) appropriate and clearly defined? (yes, no, or unclear)
	(18) Were the figures (images, illustrations, diagrams, etc.) presented in the study clear and understandable? (yes, no, or unclear)
	(19) Were any ambiguous anatomical observations (i.e., those likely to be classified as “others”) clearly described/depicted? (yes, no, or unclear)
	(20) Could the description of anatomy have in any way introduced bias into the study? (yes, no, or unclear)
Domain 5: reporting of results.	(21) Was the statistical analysis appropriate? (yes, no, or unclear)
	(22) Are the reported results as presented in the study clear and comprehensible, and are the reported values consistent throughout the manuscript? (yes, no, unclear)
	(23) Do the reported numbers or results always correspond to the number of subjects in the study? If not, do the authors clearly explain the reason(s) for subject exclusion? (yes, no, or unclear)
	(24) Are all potential confounders reported in the study, and subsequently measured and evaluated, if appropriate? (yes, no, or unclear)
	(25) Could the reporting of results have in any way introduced bias into the study? (yes, no, or unclear)

**Table 2 diagnostics-15-01858-t002:** JBI assessment tool description.

Questions	Yes	No	Unclear	Not Applicable
1. Were the patient’s demographic characteristics clearly described?				
2. Was the patient’s history clearly described and presented as a timeline?				
3. Was the patient’s current clinical condition at the time of presentation clearly described?				
4. Were the diagnostic tests or assessment methods and results clearly described?				
5. Were the intervention(s) or treatment procedure clearly described?				
6. Was the clinical condition following the intervention clearly described?				
7. Were adverse events (harms) or unanticipated events identified and described?				
8. Does the case report provide learned lessons?				
1. Assess each article with eight questions and select the answers: “yes,” “unclear,” “no,” or “not applicable.”				
2. The articles were assessed using the following criteria:				
Low risk of bias: more than 70% “yes” answers.				
Moderate risk of bias: between 50% and 69% “yes” answers.				
High risk of bias: less than 49% “yes” answers.				

**Table 3 diagnostics-15-01858-t003:** Summary of the included articles.

Author, Year	Total Number of Subjects	Age or Mean Age (SD) and/or Range	Ethicity	Country	Type of Study	Study Technique	Sex Sample
Buena et al., 2023 [3]	108	66.5 ± 17.5 years 51.5 ± 30.5 years	Caucasian	Romania	Case report	Computed tomography angiographies (CTA)	48 males 60 females
Cigali et al., 2008 [13]	1	86 years	Caucasian	Turkey	Case report	Cadaveric The anatomical dissection	1 Male
Esen et al., 2018 [14]	640	62.4 ± 13.8 years 59.1 ± 15.7 years	Not specified	Japan	Cross-sectional study	CT Angiography images.	379 Males 261 Females
González-Castillo et al., 2018 [15]	1	91 years	Not specified	Spain	Case report	Cadaveric The anatomical dissection	1 Female
Jelev and Surchev, 2001 [16]	1	67 years	Not specified	Bulgary	Case report	Cadaveric	1 male
Lovasava et al., 2017 [17]	1	72 years	Not specified	Slovak Republic	Case report	Cadaveric	1 Female
Mariolis-Sapsakos et al., 2014 [18]	1	56	Not specified	Greece	Case report	Thyroidectomy	1 Male
Moriggl and Sturm 1996 [19]	1	89 years old	Not specified	Austria	Case report	Cadaveric	Female
Natsis et al., 2009 [20]	1	72 years old	Caucasian	Greece	Case report	Cadaveric	male
Ngo Nyeki et al., 2016 [21]	1	46	Not specified	Switzerland	Case report	Thyroidectomy	1 Male
Novakov and Delchev, 2023 [22]	2	61 85	Not specified	Bulgary	Case report	Cadaveric	1 Male 1 Female
Özgüner G and Sulak O. 2014 [23]	200	9 to 40 weeks	Not specified	Turkey	Cross-sectional study	Cadaveric	100 Males 100 Females
Ray et al., 2012 [7]	25	50–60 year	not specified	India	Case report	Cadaveric	13 Males 12 Females
Sherman andand Colborn, 2003 [24]	1	Not specified	Not specified	USA	Case report	Cadaveric	1 Male
V. Archibong et al., 2023 [12]	1	54 years	Not specified	Ruanda	Case report	Cadaveric	1 male
Westrych et al., 2022 [25]	1	74 years	not specified	Poland	case report	Cadaveric	1 Female
Yang et al., 2024 [26]	1	60 years	not specified	China	Case report	Thyroidectomy	1 Female
Yilmaz et al., 1993 [27]	130	Not specified	Not specified	Turkey	Case report	Cadaveric	Not specified
Yohannan et al., 2019 [28]	1	60	South asian	India	Case report	Cadaveric	1 Male

**Table 4 diagnostics-15-01858-t004:** Risk of bias assessment according to the JBI critical appraisal check-list.

Author	JBI Q1	JBI Q2	JBI Q3	JBI Q4	JBI Q5	JBI Q6	JBI Q7	JBI Q8	Bias Risk
Novakov and Delchev, 2023 [22]		N/A	N/A		N/A	N/A	N/A		High
Sherman and Colborn, 2003 [24]		N/A	N/A		N/A	N/A	N/A		High
Yang et al., 2024 [26]							N/A		Low
Westrych et al., 2022 [25]		N/A	N/A		N/A	N/A	N/A		High
Yohannan et al., 2019 [28]		N/A	N/A		N/A	N/A	N/A		High
Yilmaz et al., 1993 [27]		N/A	N/A		N/A	N/A	N/A		High
Bunea et al., 2023 [3]		N/A				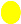	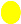		Moderate
González-Castillo et al., 2018 [15]		N/A	N/A			N/A			High
Ngo Nyeki et al., 2016 [21]									Low
Cigali et al., 2008 [13]		N/A	N/A			N/A			High
Natsis et al., 2009 [20]		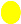	N/A	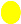		N/A	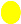		High
Moriggl and Sturn, 1996 [19]			N/A			N/A	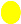		High
Jelev and Surchev, 2001 [16]		N/A	N/A			N/A			High
Mariolis- Sapsakos et al., 2014 [18]				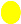					Low
V. Archibong et al., 2023 [12]			N/A			N/A			High
Lovasova et al., 2017 [17]			N/A			N/A			Moderate

Note: Green = yes; Red: no; Yellow: Unclear; N/A: not applicable.

**Table 5 diagnostics-15-01858-t005:** Types of variations of the ITA found in the included studies.

Author, Year	Absence of ITA	Differences in Origin	Additional Arteries
Archibong et al., 2023 [12]	-	-	An additional one from the left CCA
Bunea et al., 2023 [3]	-	ITA originating from the right and left SCA, left vertebral artery, and CCA	-
Cigali et al., 2008 [13]	-	-	Accessory inferior thyroid artery from the suprascapular artery
Esen et al., 2018 [14]	Multiple cases of absence without specified compensation	ITA originating from the right and left SCA, left vertebral artery, and CCA	-
González-Castillo et al., 2018 [15]	Absence of right IA in a female with situs inversus totalis, compensated by two left ITAs	-	-
Jelev and Surchev, 2001 [16]	Absence on the right side in a male, compensated by a large STA	-	-
Lovasova et al., 2017 [17]	-	ITA originating from the BCT	Three additional arteries on the right side, including MTA and aberrant accessory thyroid artery
Mariolis-Sapsakos et al., 2014 [18]	-	ITA originating from the right and left SCA, left vertebral artery, and CCA	-
Moriggl and Saturn, 1996 [19]	Bilateral absence in a female, compensated by a branch of the left internal thoracic artery	-	-
Novakov and Delchev, 2023 [22]	Bilateral absence in a female, compensated by large STAs and an ITA from the BCT	-	Right thymothyroid artery supplying both thyroid and thymus
Natsis et al., 2009 [20]	Bilateral absence of the ITA compensated by the left STA	ITA originating from the right and left SCA, left vertebral artery, and CCA	-
Ngo Nyeki et al., 2016 [21]	-	ITA originating from the right and left SCA, left vertebral artery, and CCA	-
Özgüner and Sulak, 2014 [23]	-	ITA originating from the right and left SCA, left vertebral artery, and CCA	-
Ray et al., 2012 [7]	-	ITA originating from the right CCA supplying the left thyroid lobe	-
Sherman and Colborn, 2003 [24]	Absence on the left side in a male, with compensation by right ITA branches	-	-
Westrych et al., 2022 [25]	-	-	Common trunk for ITA, internal thoracic artery, and thyrocervical trunk
Yang et al., 2024 [26]	-	ITA originating from the right and left SCA, left vertebral artery, and CCA	-
Yilmaz et al., 1993 [27]	Bilateral absence in an unspecified case, compensated by an ITA from the left BCT	-	-
Yohannan et al., 2019 [28]	Bilateral absence in a male, compensated by an ITA from the subclavian artery	-	-

**Table 6 diagnostics-15-01858-t006:** ITA’s morphological description and clinical correlations of the included articles.

Author, Year	Inferior Thyroid Artery Variation and Other Variations	Clinical Consideration
Buena et al., 2023 [3]	108 ITAs, 31.48% (*n* = 34) originated directly from the subclavian artery (SCA), while 68.52% (*n* = 74) arose from the thyrocervical trunk (TCT). Laterally, right-sided ITAs demonstrated SCA origins in 31.25% (20/64) of cases versus 31.82% (14/44) on the left. Gender analysis revealed males exhibited higher SCA origins (37.50%, 18/48) compared to females (26.67%, 16/60), with right-sided male ITAs more frequently originating from the SCA (55.56%, 10/18) than left-sided (44.44%, 8/18). TCT origins predominated across both sexes (males: 62.50%, females: 73.33%), particularly on the right side in females (63.64%, 28/44). These variations are consistent with documented anatomical diversity in cervical vasculature.	-
Cigali et al., 2008 [13]	An unusual unilateral variation was observed where the left ITA originated anomalously. Alongside the standard inferior and STA, a third thyroid artery (identified as an accessory ITA) arose from the left suprascapular artery. This accessory artery descended ~1 cm vertically before branching medially, coursing anterior to the carotid sheath, phrenic nerve, and internal thoracic artery, and terminating at the left inferior thyroid pole. The inferior thyroid artery maintained its typical branching pattern, giving rise to the ascending cervical artery. This variant highlights rare collateral thyroid vasculature, which could influence surgical planning due to its proximity to critical neurovascular structures	An accurate knowledge of the normal anatomy of the thyroid gland vessels, particularly of their patterns of variation, is important in parathyroid localization studies, and in neck surgery procedures. especially in tracheostomy.
Esen et al., 2018 [14]	The right and left superior thyroid arteries arose from the external carotid artery in 413 (64.5%) and 254 (39.7%) patients, from the bifurcation of the common carotid artery in 131 (20.5%) and 148 (23.1%) patients, and from the common carotid artery in 90 (14.1%) and 226 (35.3%) patients, respectively. We could not observe the right and the left superior arteries in 6 (0.9%) and 12 (1.9%) of the patients, respectively. However, the right and left inferior thyroid arteries thyroid were not identified in 14 (2.2%) and 45 (7%) of the patients, respectively. The thyroidea ima artery was detected in 2.3% of the patients.	The high variability of the origin and the occur rence rate of the TIA can lead to bleeding during surgery or tracheostomy
González-Castillo et al., 2018 [15]	A description is made of how situs inversus totalis occurs in the abdomen and thorax, therefore, not so much detail is given on the inferior thyroid artery. It is noted that a double left inferior thyroid artery and agenesis of the left adrenal gland were observed—two variants that have not been previously reported in association with situs inversus..	The thyroid primordium is initially irrigated by a dense arterial network, including lateral branches from the subclavian arteries. Post-development, this network typically regresses, leaving the four principal thyroid arteries as the dominant supply. Persistent embryonic vessels, such as an accessory inferior thyroid artery arising from the thyrocervical trunk, may remain. Inferior thyroid artery ligation is critical during thyroidectomy; undetected variants risk intraoperative hemorrhage.
Jelev and Surchev, 2001 [16]	In the male specimen, a right thymothyroid artery originated 48 mm proximal to the carotid bifurcation (CCA diameter: 8 mm), dividing into three thyroid branches (middle branch: 63 mm length, 2.5 mm diameter) and a thymic branch (1.5 mm diameter), while the left ITA atypically arose from the CCA (50 mm from bifurcation; diameter: 2.5 mm) with superior (10 mm, 1.5 mm) and inferior (16 mm, 0.8 mm) branches. In the female, bilateral ITA absence was compensated by hypertrophied STAs and a TIA from the brachiocephalic trunk (28 mm from origin; 4 mm diameter), bifurcating into thymic and ITA branches (right: 3 mm trunk; left: 2 mm ITA with isthmic branch). These variations, including accessory thymothyroid arteries and TIA-mediated compensation, highlight embryological persistence of aortic arch derivatives and necessitate preoperative vascular mapping to avoid hemorrhage during cervical surgeries.	The thyroid gland’s rich vascularization makes the dissection of its vessels a crucial aspect of thyroid surgeries. While the STAs tend to have stable anatomical positions, the inferior thyroid arteries exhibit significant variability. The recurrent laryngeal nerve’s proximity to the inferior thyroid artery is particularly concerning during thyroidectomies, as this area is considered highly vulnerable due to complex nerve-artery relationships. Thus, understanding the anatomical variations of the ITA is essential for minimizing surgical complications.
Lovasava et al., 2017 [17]	The thyroid gland is typically supplied by the STA and ITA, with occasional anatomical variations such as the TIA, an inconstant vessel originating from the brachiocephalic trunk or aortic arch. Vascular anomalies like the TIA, which ascends to the thyroid isthmus, and rare branches such as MTAs or AITAs arising from the common carotid artery, highlight the diversity in thyroid vasculature. These variations, alongside broader vascular anomalies (e.g., lingual artery course deviations or persistent embryonic arteries), emphasize the importance of preoperative imaging to avoid complications during neck surgeries.	The risk of blood vessel damage during the surgery can be minimized by keeping in mind all the anatomical variations and development anomalies. In the area of the neck, the anterior cervical region is the most important one from clinical point of view and the area of the thyroid gland must be evaluated precisely before any surgical procedures.
Mariolis-Sapsakos et al., 2014 [18]	The lower lobe of the thyroid was found to be supplied by left and right inferior thyroid arteries, which originated bilaterally from the common carotid artery.	Rare variation during procedures in the neck area, and prior knowledge will be beneficial to limit the incidence of possible complications
Moriggl and Sturm 1996 [19]	Both inferior thyroid arteries and the left superior thyroid artery were absent. The right superior thyroid artery originated from the terminal right common carotid artery (RCA) to supply the isthmus. The left internal thoracic artery (ITA) exhibited an enlarged proximal segment, emitting ascending cervical and suprascapular arteries before bifurcating at the first intercostal space: a dominant lowest thyroid artery ascending to the thyroid’s inferior margin and a standard ITA continuation. The lowest thyroid artery bifurcated midline into twin branches (equal caliber) that traversed the trachea anteriorly to perfuse the thyroid lobes.	All diagnostic and/or surgical procesdures involving supraesternal fossa, tracheostomy in particular, require a carfeul approach because of a possibly existing lower thyroid artery. Once cut or injured it can cause extensive and uncontrollable bleeding. This is especially true if this vessel is huge. Finally, description of arterial variation, especially if they are of rare occurrence, is important for interpretation within the scope of modern imaging techniques.
Natsis et al., 2009 [20]	Left vertebral artery originated from the root of the left subclavian artery, close to the aortic arch. Following its route upwards to the neck we came across another arterial branch that arose from it having and upward to the midline course, that was the inferior thyroid artery.	It should be known by neck surgeons in order to avoid implication during thyroidectomy while trying to ligate the regional vessels. Vascular interventionalist and angiographers should also bear in mind this variation during inferior thyroid artery catherization either diagnostic or therapeutic, in case of an aneurysm or a rupture at the thyroid region.
Ngo Nyeki et al., 2016 [21]	The right inferior thyroid artery (ITA) was originating directly from the right common carotid artery, right ITA moved towards the lower third of the right thyroid lobe where it divided into three branches. The contralateral ITA originated from the thyrocervical trunk and the superior thyroid arteries from the external carotid arteries, as expected.	Two major risks from a surgical perspective: the first one is hemorrhagic by injuring this artery when on aberrant course; the second risk is increased injury to the recurrent laryngeal nerves (RLN).
Novakov and Delchev, 2023 [22]	In the male specimen, a right thymothyroid artery was noted to originate 48 mm from the carotid bifurcation and branched into thyroid and thymic branches. The left inferior thyroid artery (ITA) arose unusually from the common carotid artery, with distinct superior and inferior branches. In the female specimen, the absence of bilateral ITAs was compensated by enlarged superior thyroid arteries and a thyroidea ima artery from the brachiocephalic trunk. These anatomical variations highlight the importance of preoperative vascular mapping, such as CT angiography, to reduce surgical risks during thyroidectomies or mediastinal procedures.	-
Özgüner G and Sulak O. 2014 [23]	The origins of the ITA were as follows: thyrocervical trunk in 97.5% of cases on the right and 96.5% on the left; CCA in 2.5% of cases on the right and 2.5% on the left. The ITA was absent on the left side in two cases. The location of the ITA in relation to the entrance of the thyroid lobe was determined and three different types were detected: 1. The ITA entered the thyroid lobe from the subpolar; right 79 (39.5%), left 78 (39%). 2. “The ITA entered the thyroid lobe from the lower one-third aspect.”; right 101 (50.5%), left 102 (51%). 3. The ITA entered the thyroid lobe from the one-third middle side; right 20 (10%), left 18 (9%).	The relationship of the RLN to the ITA is highly variable, and the variability of the ITA and its position relative to the RLN makes it a poor surgical landmark.
Ray et al., 2012 [7]	Origin branching pattern: In the present study, ITA arose as a single trunk from TCT in 5 males and 7 females and as multiple branches it was observed in 8 males and 5 females. The incidence of multiplication is more in the males. Branching patter: The mean number of branches from STA and ITA was found to be three Lenght: Longer arteries were observed on left side.	Preservation of all the branches is important during thyroidectomy while ligating the regional vessels (31). The variation should also be borne in mind during ITA catheterization for diagnostic or therapeutic purposes in case of an aneurysm.
Sherman and Colborn, 2003 [24]	The case report describes a cadaver with an atypically small ITA that followed its normal trajectory but branched into several small vessels, ultimately not supplying the thyroid gland. In contrast, the right inferior thyroid artery divided into two branches that provided blood to the inferior pole of the right lobe, the thyroid isthmus, and the inferior pole of the left lobe. This incomplete development of the ITA disrupted the typical anatomical relationship with the left recurrent laryngeal nerve, potentially increasing the risk of nerve injury during surgical procedures.	As an alternative surgical approach, ligation of the tertiary branches of the inferior thyroid artery at the surface of the thyroid gland, with subsequent identification of the recurrent laryngeal nerve, can aid in protecting the nerve
V. Archibong et al., 2023 [12]	The ITA usually originates from the thyrocervical trunk in the majority of the human population (90.5%), or from the subclavian artery in a few populations of humans (7.5%). It is quite rare to find the ITA originating from the CCA. The case was a male who had two inferior thyroid arteries on the left side, with one originating from the TCT and the other taking a rare anatomical origin from the CCA. The ITA on the right side was one and from the normal source.	Documenting anatomical arterial patterns is crucial for increasing awareness among surgeons and imaging specialists, thereby minimizing the risk of iatrogenic complications during thyroid surgeries. The ITA’s close relationship with the recurrent laryngeal nerve, which is vital for vocal cord mobility and respiratory functions, underscores the importance of visualizing this anatomical relationship to ensure nerve preservation during surgical procedures.
Westrych et al., 2022 [25]	The ITHA originated from the common trunk. The ITHA enters the thyroid gland and anastomoses with the contralateral ITHA and ipsilateral superior thyroid artery.	ITHA: Thyroidectomy is a common procedure in general and endocrine surgery in which ligation of the ITHA is performed. Therefore, the correct location of the ITHA is key to a well-performed surgery Common trunk: “This variation may result in the failure of coronary bypass surgery due to a coronary steal associated with a common ITA-ITHA-TCT trunk, highlighting the importance of considering its potential causes.”
Yang et al., 2024 [26]	The inferior thyroid artery, which exhibited a diameter of approximately 0.5 cm and originated directly from the BCT. The distance between inferior thyroid artery and aortic arch was about 1.5 cm. The inferior thyroid artery originating from the BCT was approximately 2 cm long and divided into two branches to supply blood for the thyroid gland.	-
Yilmaz et al., 1993 [27]	ITA: there was found the absence of both inferior thyroid arteries. IMA: An IMA artery arose from the brachiocephalic trunk as a short, wide vessel. It bifurcated almost immediately into two branches, reaching the bases of both the right and left thyroid gland lobes. Subclavian Artery: The first part of the left subclavian artery is anomalous due to the absence of the thyrocervical trunk, costocervical trunk, and vertebral artery. Vertebral artery: The left vertebral artery arises from the aortic arch between the left common carotid artery and the left subclavian artery.	Authors note: “An accurate knowledge of the normal anatomy of the thyroid gland vessels, and particularly of their patterns of variation, is important for parathyroid localisation studies, in neck surgery procedures and especially in tracheostomy”
Yohannan et al., 2019 [28]	ITA: There was absence of the inferior thyroid artery on the right and left side. IMA: The ima artery originated from the subclavian artery, close to the origin of the vertebral artery. The ima artery was seen to take an anterior course between the common carotid artery medially and the internal jugular vein and the vagus nerve laterally. The artery then traced a path medially, superficial to the common carotid artery to reach the lower pole of the right lobe of the thyroid	The thyroidea ima artery’s anterior tracheal position increases hemorrhage risk during lower neck procedures (e.g., tracheostomy, laryngeal transplantation). Though typically superficial and oblique in the neck, its terminal branches may interface with tracheal structures. “Surgeons must remain highly aware of this anatomical variant to prevent inadvertent vascular injury.”

## Data Availability

Not applicable.

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
