# Peer review of "A Systematic Review of Anatomical Variations of the Inferior Thyroid Artery: Clinical and Surgical Considerations"

_diagnostics, 2025, doi:10.3390/diagnostics15151858_

Round 1

Reviewer 1 Report

Comments and Suggestions for Authors

This is an interesting contribution trying to summarize a number of publications on the anatomy of the inferior thyroid artery (ITA) in man.  The authors applied stringent quantitative analytical criteria for their study, focused on the consistency of their conclusions with the quality of the selected publications, and raised a number of adequate considerations on the clinical impact of the parametric distributions they obtained. However, it would be advisable to provide some specifications and clarifications to ameliorate and substantiate the work:    

ABSTRACTS

The authors state “A comprehensive search was conducted in multiple databases from inception to November 2024”.  Thus, apprently they proceeded through some form of meta-analysis, otherwise it would be quite difficult to drive conclusions on the reliability of data from a wide literature on the topic chosen.  In this case, and also in any other approach discussed, it should be clearly summarized the time interval of literature search, avoiding generical statements like “from inception”.  What would be the date the authors started they search?  It is a critical and important point due to the changes throught years in methodologies for recording anatomical data on vessels, leading to many false negative and positives to be taken into account.  So, please, specify the time interval of search, and list the library or electronic/web sources they used to select articles entering the study.

In addition, it is not clear to the reviewer what the authors mean for:”correlation with pathologies”. Which correlation? What type of pathology?  Please, clarify.

INTRODUCTION

Correctly, the authors point out a number of specific advantages in the knowledge of the ITA anatomy, primarily for the surgical approach to the thyroid gland.  However, a certain attitude to repetition of concepts already expressed emerges, making the Introduction quite long.  The reviewer suggests to summarize the different perspectives, and provide a more stringent Introduction.

METHODS

In the section on Eligibility Criteria the authors mention as outcomes the ITA variants.  However, they do not specify which type of variants they collected/studied (number of vessels, origin, anastomoses, angles of trajectory, etc…).  Please, specify this point.

The authors also mention as an outcome the correlation of ITA with pathologies.  To the reviewer, this is unclear:  what did the authors mean?  and, which pathologies?  Please, clarify.

RESULTS

The reviewer does not clearly understand the following passage “and after the duplicate  records were removed and the records removed by automation tools and other reasons.” Is just a problem of reorganizing the english language or of meaning of the procedure applied?  Please, clarify the statement (including “other reasons”, Which reasons?).

At the beginning of the section of the Methological quality assessment (lines 200-219), please reformulate the english language to provide a more fluent statement.

Slightly below, the authors mention domains 1, 3, 5. What type of domains are those?  Please, clarify.  

DISCUSSION

The authors start presenting a “systematic review” with a fairly high number of data.  Though authors correctly checked with the AQUA and JBI techniques the reliability of their informations, to provide robust conclusions on the clinical usefulness to their study it would have been adequate to have a semiquantitative or quantitative meta-analytical approach comparing the strenght of different data on each variational parameter between studies.  Thus, at this point the reviewer suggest to include a short opening perspective on this weakness, possibly immediately merging with section 4.1 inherent the other methodological considerations.  

The reviewer appreciate the enlightment on ITA embryology, and its possible multiple sources of derivation in relation to different germ layers.  However, it is not clear to him the reason why the authors introduced this, apparently without any connection to their data (though a very short but generical passage later in the discussion).  Please, try to give more consequentiality to the reasoning linking complex ITA embryology and quantitative evidence of the parameters the author record in their study.

In section 4.1 the authors again introduce the concept of the analytical “domains” their considered but, as already remarked in the Results section, they do not provide any specification on these domains beyond their own acquaitance of the techniques used for assessing relevance of their data.  Please, make at this stage a short note on these concepts, and clarify to the reader the meaning of these “domains”, including importance for the conclusions they like to reach.

Collectively, the authors remain quite generical in discussing the role of the ITA and its variational anatomy in relation to the different surgical techniques currently used.  In particular, they do not tackle the challange of the most modern robotic-assisted endoscopic approaches including trans-axillary, bilateral areola, and trans-oral methodologies.  Indeed, all these may introduce additional surgical difficulties in relation to anatomical variants of the ITA. Please, make a note on this. Similar, the reviewer would recommend the authors to mention the diagnostic role of the ITA in thyroid ultrasound, and its role in clinical follow-up of nodular goiters.

FIGURES AND TABLES

For the reader, Table 3 is quite difficult to be followed, then losing strenght in reading.  Please, reorganize in a more concise and readable manner, avoiding the long descriptions included.

A similar criticism could also be done for Table 4.  The reviewer suggest to use abbreviations in all Tables and, provide a Summary Table of abbreviations a the beginning of the manuscript.       

Comments on the Quality of English Language

The reviewer urges the authors to proceed with a reassesment of the English language of the manuscript, due to sections lacking pronouns, close repetition of the same word and/or the same concept, and so on including Tables.       

Author Response

Dear Reviewers,

We sincerely appreciate the time and effort you have dedicated to reviewing our manuscript. Your insightful comments and constructive feedback have been invaluable in refining our work and enhancing the clarity and impact of our findings. We are grateful for the opportunity to address your suggestions and provide further clarification where needed. Below, we respond to each of your comments and outline the revisions made in the manuscript.

  1. Comment on Conclusions Section: Thank you for highlighting the need to summarize the results more succinctly and focus on specific variations and their frequencies. We have revised the Conclusions section to clearly summarize the key findings regarding the ITA's variations and their clinical significance, ensuring that the conclusions drawn are directly supported by the data presented.
  1. Reevaluation of Standards: We appreciate your suggestion to emphasize the potential need for reevaluating anatomical and surgical standards. In response, we have incorporated a discussion on the implications of our findings for current clinical practices and the importance of recognizing anatomical diversity in surgical planning.
  1. Clarification and Consistency: We have carefully reviewed the manuscript for any vague statements and ensured that all information presented is clear and directly related to our study's objectives and results. Any opinions or interpretations have been confined to the Discussion section, as per your advice.

We trust that these revisions meet your expectations and further strengthen the manuscript. Thank you once again for your valuable feedback and guidance. Please feel free to reach out for any further clarifications or additional information.

Sincerely,

Reviewer 2 Report

Comments and Suggestions for Authors

Dear authors,

It has been an honor and a pleasure to act as a referee for your manuscript ‘Anatomical Variations of the Inferior Thyroid Artery: Clinical and Surgical Considerations.’, which revisits an interesting question with a potential direct application in daily clinical practice. I have learned after reading it, which is one of the best compliments to the work of a colleague. However, although the study in itself is well performed, the way in which results are presented makes the manuscript very unfriendly to the reader and blurs its results. Also relevant is the fact that this review adds little information to the field, as most of this work was performed by Branca et al., 2022.

The main general problem is that the text is incredibly reiterative, spinning around the same concepts over and over and again. It is necessary to state only once or twice the relevance of the anatomic variations of the arteries, and it is not necessary to reiterate the level of quality of each study in every section of the manuscript; a couple of supplementary tables explaining the criteria and the figure that is already included are more than enough. Also, the rationale underlying a review is to sort information together under a single coherent stream: the way in which results are written, making a summary of each paper, is not very useful for readers and makes the text longer and reiterative. Finally, I would prefer to see the results extracted from single cases separated in the form the results extracted from cross-sectional studies, as both kinds of studies serve different purposes. (The first one informs of the kind of variations and the second ones provide information about the prevalence of variations). I would also wish to see references cited in a coherent way, as this present state provides the impression of an uncareful manuscript preparation. Below, authors will find more detailed comments in a section-wise manner:

Introduction

Very reiterative, while vague at some points. It relies excessively on a pair of references that are cited at every step.  The paragraph in lines 86-94, that frames the clinical problem, should be the beginning one. In a more specific way:

Line 51-52: too vague; how does it work as a marker? Why is it controversial?

Line 60-63: authors should choose between deepening the evolutionary perspective, or scrapping this paragraph altogether; its present form adds vagueness without providing insights

Lines 77-79: this paragraph is irrelevant

Lines 97-99: this paragraph is irrelevant

Lines 100-106: very redundant

Methods

Generally good. However, the search strategy should be more detailed, not only in the supplementary files but also in the main text. The methods included in lines 163-171 deserve to be illustrated in a table or a schematic figure.

Results.

Authors deserve to be complimented on the quality of the Figure 4. It might be improved by highlighting (colouring) the ITA in each case to draw attention to it. However, the way in which this figure is referenced in the text is somewhat uncareful as it is mentioned only a couple of times and without details. It would be better for the readers to be directed to a subsection of the figure (eg. Fig. 4G, and so on…), whenever a specific variation is mentioned. Section 3.3 is poorly written and complicated to understand, please simplify and provide a table with the ‘questions’ you apply to the text. Section 3.5 is probably unnecessary as part of the outcomes, and more relevant in the discussion section. Finally, Table 1 duplicates information from the text. There are some details, like the sex of the case, that interrupt the flow of the reading and are better recorded in a table (is it necessary to state in the middle of the text an unspecified gender like in line 274, 315). Table 3 is too complicated, please homogenize the way in which the results are presented in the columns, and be simpler, the way in which this table is elaborated is contradictory with the concept of table, that should be a rapid way to find results and not another (and more uncomfortable) part of the text. Finally, I bear some doubts about the pertinence of the study of González-Castillo et al., 2018: a situs inversus is not normal anatomy and any variation found at this level should be considered as part of the polimalformative sequence.

Discussion

Too long, and too reiterative. There are many unimportant assertions (e.g line 368), paragraphs that almost duplicate information (lines 480-487 and 489-495).

More specific comments:

Lines 379: I do not clearly see the ethnic variability in these references (please, state more clearly). None of the studies stratify by ethnicity, and therefore it is very difficult to state that the variation in the numbers is due to interethnic differences (rather than, for example, regional variants). Also, the immense majority of the papers you cite are case reports that provide no epidemiological information.

Lines 385-386: this part could be interesting, but it is written too vaguely

Line 393: Please, be careful with the text and the way you cite. It appears that there is a deleted sentence.

Lines 401-402: please, references

Lines 421-424: unnecessary

Lines 432-439: this paragraph is, on one hand, reiterative and on the other hand lacking references or even figures, for a potentially relevant question. This happens again in lines 469-472, where a very relevant comment is lost in the middle of iterations of the same ideas and not enough developed.

Conclusions:

Please, summarize your results (the different kinds of variations and their frequences) and do not repeat vague statements.

Overall, a well-designed study that (however) lacks novelty and results too reiterative in the way in which is written. I hope authors find the comments useful.

Un saludo

Author Response

Dear Reviewers,

We sincerely appreciate the time and effort you have dedicated to reviewing our manuscript. Your insightful comments and constructive feedback have been invaluable in refining our work and enhancing the clarity and impact of our findings. We are grateful for the opportunity to address your suggestions and provide further clarification where needed. Below, we respond to each of your comments and outline the revisions made in the manuscript.

  1. Comment on Conclusions Section: Thank you for highlighting the need to summarize the results more succinctly and focus on specific variations and their frequencies. We have revised the Conclusions section to clearly summarize the key findings regarding the ITA's variations and their clinical significance, ensuring that the conclusions drawn are directly supported by the data presented.
  1. Reevaluation of Standards: We appreciate your suggestion to emphasize the potential need for reevaluating anatomical and surgical standards. In response, we have incorporated a discussion on the implications of our findings for current clinical practices and the importance of recognizing anatomical diversity in surgical planning.
  1. Clarification and Consistency: We have carefully reviewed the manuscript for any vague statements and ensured that all information presented is clear and directly related to our study's objectives and results. Any opinions or interpretations have been confined to the Discussion section, as per your advice.

We trust that these revisions meet your expectations and further strengthen the manuscript. Thank you once again for your valuable feedback and guidance. Please feel free to reach out for any further clarifications or additional information.

Sincerely,

M.Orellana

Reviewer 3 Report

Comments and Suggestions for Authors

The work aroused great interest in me because, as a head and neck surgeon and an anatomy scholar (I am a professor), I believe it is an important subject.
By reading the work I will try to contribute.
Some statements I have to put about criticism

in the introduction:
"the main importance lies in its role as a useful, although highly controversial, marker for accessing the cervical region" this information has an opinion, I would prefer that opinions come into the discussion and be discussed

Some inaccuracies in the introduction:

"The data collected in the scientific literature in relation to anatomical variants should be compiled and published, the visibility of this evidence is relevant both from the clinical and surgical point of view. On this latter point, literature highlights that between the peri-thyroid sheath and the capsule that adheres to the parenchyma, the so-called “dangerous space” must remain intact when the gland is removed, since here it houses the rich thyroid venous network. While also considering the possibility of nerve laceration, it is worth noting the recurrent laryngeal nerve, which is closest to the left lobe of the gland.(Branca et al., 2022)" 
"the peri- thyroid sheath and the capsule that adheres to the parenchyma, the so-called “dangerous space” must remain intact when the gland is removed"What would this perithyroid sheath be, would it be the medial lamina of the cervical fascia? In an anatomical work I cannot accept statements like this, besides this paragraph seems disconnected to me.

"Thyroidectomy is a fairly frequent surgical intervention, however, the anatomical structures of the region such as the relationship of the RLN with the ITA make the procedure very complex. The lesion of the RLN is one of the main complications of thyroidectomy, which occurs in 2-5% of patients without nerve variations, and as a consequence generates voice and swallowing problems"This statement is also subject to criticism. After 2 to 5% of patients, vocal fold paralysis is not always due to nerve injury and only to its manipulation, even if careful. I would prefer to review this issue

In my opinion, the introduction still lacks a clear objective of the work and a focused answer in the conclusion (the conclusion is verbose).

The methods make it clear how you did it (PRISMA), but your inclusion criteria leave me confused, case reports? How many cases, and then you talk about deviations and attribute neglect to cadaver dissection work? Why? I would suggest clarifying the inclusion criteria, as are the exclusion criteria below, and explaining what was considered deviations from the studies, because this escaped me when I read it
"1) population: sample of dissections or images of the ITA; 2) outcomes: ITA prevalence, variants, and their correlation with pathologies. Additionally, anatomical variants were classified and
described based on normal anatomy and classifications proposed in literature; 3) studies: this systematic review included research articles, research reports, or original research published in English or Spanish in peer-reviewed journals and indexed in some of the databases reviewed. Conversely, the exclusion criteria were as follows: 1) population: animal studies; studies: letters to the editor or comments."

This statement with the abbreviations is also unclear: "Two authors (CP and TP)"???

Your  results presented in monographic form are difficult to understand, the tables need to be presented in a more concise way, they are very descriptive... perhaps to group patterns or qualify studies

In your discussion we have again statements that I disagree with:

"Studies indicated that variations in the ITA's trajectory and branching could lead to surgical risks, including uncontrollable hemorrhage during procedures such as tracheotomy Moriggl & Sturn, 1996; Ray et al., 2012"
Tracheotomy reaches little or nothing in the inferior thyroid artery

"internal mammary artery" what is this area would it be the internal thoracic artery? “During thyroidectomy, ligation of the inferior thyroid artery is necessary, so the presence of an accessory artery that is not known about at the time of surgery can cause unexpected bleeding during the procedure (González-Castillo et al 2018). in case of aneurysms or rupture of arteries in the thyroid region (Natsis et al 2009)”, again an attribution in a context that I think is remote.

"In summary, the variations of the ITA are significant not only for anatomical under standing but also for their clinical implications in surgical practice. The high incidence of anatomical variations in the ITA necessitates that surgeons remain vigilant during procedures involving the neck, such as thyroidectomies and tracheotomies” This is a well-placed opinion but it loses context with the previous statements.

Second, a significant proportion of the studies exhibited a high risk of bias, particularly those based on cadaveric dissections” here I would need to understand ...

Finally, as I said, your conclusions are not focused
Think about reevaluating standards and importance, make sure that opinions come up in the discussion and finally be aware that some information is taken into account

Author Response

Dear Reviewers,

We sincerely appreciate the time and effort you have dedicated to reviewing our manuscript. Your insightful comments and constructive feedback have been invaluable in refining our work and enhancing the clarity and impact of our findings. We are grateful for the opportunity to address your suggestions and provide further clarification where needed. Below, we respond to each of your comments and outline the revisions made in the manuscript.

  1. Comment on Conclusions Section: Thank you for highlighting the need to summarize the results more succinctly and focus on specific variations and their frequencies. We have revised the Conclusions section to clearly summarize the key findings regarding the ITA's variations and their clinical significance, ensuring that the conclusions drawn are directly supported by the data presented.
  1. Reevaluation of Standards: We appreciate your suggestion to emphasize the potential need for reevaluating anatomical and surgical standards. In response, we have incorporated a discussion on the implications of our findings for current clinical practices and the importance of recognizing anatomical diversity in surgical planning.
  1. Clarification and Consistency: We have carefully reviewed the manuscript for any vague statements and ensured that all information presented is clear and directly related to our study's objectives and results. Any opinions or interpretations have been confined to the Discussion section, as per your advice.

We trust that these revisions meet your expectations and further strengthen the manuscript. Thank you once again for your valuable feedback and guidance. Please feel free to reach out for any further clarifications or additional information.

Sincerely,

M. Orellana

Round 2

Reviewer 1 Report

Comments and Suggestions for Authors

The reviewer agrees with all the changes and ameliorations the authors made to their manuscript.  As a result of their work in providing a sketch of the only possibly classical meta-analytical approach they obtained, the reviewer suggest they introduce the plot as supplementary material, with a recall of it in the Discussion

Author Response

Dear Reviewers,

We are truly grateful for your continued engagement and insightful feedback on our manuscript, "Anatomical Variations of the Inferior Thyroid Artery: Clinical and Surgical Considerations." Your contributions have been invaluable in refining our work to ensure clarity and scientific rigor. We are pleased to submit this revised version, addressing all your suggestions and concerns. Your detailed reviews have significantly enhanced the quality of our manuscript. We have carefully addressed each point raised, aiming to meet the high standards expected.

Sincerely,

Mathias Orellana.

Reviewer 2 Report

Comments and Suggestions for Authors

Dear authors, 
it has been an honour to read your revised version of the manuscript  'Anatomical Variations of the Inferior Thyroid Artery: Clinical and Surgical Considerations.'. This new version clearly improves the readability of your work. Still, there are many unaswered questions remaining: 

1. It still not clear which information is added to the field, particularly compared to some reviews you already cite in the introduction. This should be clearly commented in the last sentences of the introduction. 
2. There is still too much attention paid (e.g lines 167-185 of material and methods and lines 872-972 in the discussion) to methodological checklists of the studies included in your review compared to their results, which should be the focus. 
3. The sub-chapter 3.4 is the backbone of your work. Although improved, it still lacks clarity (there is no need to mention the authors of each study in the text, this purpose is fulfilled by the references) and specific references to your Figure 4. ALso, there are unsubstained affirmations (lines 282-284) that are more suitable for the discussion. If you mention sex differences, please, mention them directly and not as a vague statement. Also, the study of Bunea et al (2023) is not a case report. 
4. Table 5 still inlcudes too much text, is too heterogeneus, and is not helpful to the readers
5. Discussion is still too long. The newly added (thank you) embryological considerations deserves a separate chapter. 

Un saludo y suerte. 

Author Response

(The authors gave the same response as above.)

Reviewer 3 Report

Comments and Suggestions for Authors

The review with modifications brought improvement to your work: I think it can be published. There are some imperfections in the writing and points that I disagree with, but they are less relevant.
I will give an example:
"Among characteristics observed in the compiled studies about study subjects, the 212 gender distribution of the 1,118 subjects/patients recorded includes a total of 439 (39.27%) 213 females, 549 (49.11%) males." The numbers in this paragraph do not add up, but you can correct them later in the results. Make this correction in this separate paragraph. The rigor of the selection seems very clear and I congratulate you.

In the discussion, and this is an opinion, I could have more opinions from you: "The research highlighted variations in the ITA and their significant surgical implications. Studies indicated that variations in the ITA's trajectory and branching could lead to surgical risks, including uncontrollable hemorrhage during procedures such as tracheotomy Moriggl & Sturn, 1996; Ray et al., 2012" This is a statement by an author that I would refute... In my opinion, the IMA artery is what matters in tracheostomies.

Finally, I would make a more focused conclusion. There are many ITA variations, if they can be seen, that is more information. But, now follows a provocation (which does not need to be included). Nowadays, with the exams we have, is it really cost-effective to know all these variations before a surgical procedure?

My position: it is necessary to know the possibility, but I would not do a dedicated study before each thyroidectomy I perform (and I perform about 250 a year).
En

Author Response

(The authors gave the same response as above.)
